# Energy efficient synaptic plasticity

Ho Ling Li[1], Mark CW van Rossum[1,2]*

[1]School of Psychology, University of Nottingham, Nottingham, United Kingdom; [2]School of Mathematical Sciences, University of Nottingham, Nottingham, United Kingdom

**Abstract** Many aspects of the brain's design can be understood as the result of evolutionary drive toward metabolic efficiency. In addition to the energetic costs of neural computation and transmission, experimental evidence indicates that synaptic plasticity is metabolically demanding as well. As synaptic plasticity is crucial for learning, we examine how these metabolic costs enter in learning. We find that when synaptic plasticity rules are naively implemented, training neural networks requires extremely large amounts of energy when storing many patterns. We propose that this is avoided by precisely balancing labile forms of synaptic plasticity with more stable forms. This algorithm, termed synaptic caching, boosts energy efficiency manifold and can be used with any plasticity rule, including back-propagation. Our results yield a novel interpretation of the multiple forms of neural synaptic plasticity observed experimentally, including synaptic tagging and capture phenomena. Furthermore, our results are relevant for energy efficient neuromorphic designs.

*For correspondence: mark.vanrossum@nottingham.ac.uk

## Introduction

The human brain only weighs 2% of the total body mass but is responsible for 20% of resting metabolism (*Attwell and Laughlin, 2001*; *Harris et al., 2012*). The brain's energy need is believed to have shaped many aspects of its design, such as its sparse coding strategy (*Levy and Baxter, 1996*; *Lennie, 2003*), the biophysics of the mammalian action potential (*Alle et al., 2009*; *Fohlmeister, 2009*), and synaptic failure (*Levy and Baxter, 2002*; *Harris et al., 2012*). As the connections in the brain are adaptive, one can design synaptic plasticity rules that further reduce the energy required for information transmission, for instance by sparsifying connectivity (*Sacramento et al., 2015*). But in addition to the costs associated to neural information processing, experimental evidence suggests that memory formation, presumably corresponding to synaptic plasticity, is itself an energetically expensive process as well (*Mery and Kawecki, 2005*; *Plaçais and Preat, 2013*; *Jaumann et al., 2013*; *Plaçais et al., 2017*).

To estimate the amount of energy required for plasticity, *Mery and Kawecki (2005)* subjected fruit flies to associative conditioning spaced out in time, resulting in long-term memory formation. After training, the fly's food supply was cut off. Flies exposed to the conditioning died some 20% quicker than control flies, presumably due to the metabolic cost of plasticity. Likewise, fruit flies doubled their sucrose consumption during the formation of aversive long-term memory (*Plaçais et al., 2017*), while forcing starving fruit flies to form such memories reduced lifespan by 30% (*Plaçais and Preat, 2013*). A massed learning protocol, where pairings are presented rapidly after one another, leads to less permanent forms of learning that don't require protein synthesis. Notably this form of learning is energetically less costly (*Mery and Kawecki, 2005*; *Plaçais and Preat, 2013*). In rats (*Gold, 1986*) and humans (*Hall et al., 1989*, but see *Azari, 1991*) beneficial effects of glucose on memory have been reported, although the intricate regulation of energy complicates interpretation of such experiments (*Craft et al., 1994*).

Motivated by the experimental results, we analyze the metabolic energy required to form associative memories in neuronal networks. We demonstrate that traditional learning algorithms are

**eLife digest** The brain expends a lot of energy. While the organ accounts for only about 2% of a person's bodyweight, it is responsible for about 20% of our energy use at rest. Neurons use some of this energy to communicate with each other and to process information, but much of the energy is likely used to support learning. A study in fruit flies showed that insects that learned to associate two stimuli and then had their food supply cut off, died 20% earlier than untrained flies. This is thought to be because learning used up the insects' energy reserves.

If learning a single association requires so much energy, how does the brain manage to store vast amounts of data? Li and van Rossum offer an explanation based on a computer model of neural networks. The advantage of using such a model is that it is possible to control and measure conditions more precisely than in the living brain.

Analysing the model confirmed that learning many new associations requires large amounts of energy. This is particularly true if the memories must be stored with a high degree of accuracy, and if the neural network contains many stored memories already. The reason that learning consumes so much energy is that forming long-term memories requires neurons to produce new proteins. Using the computer model, Li and van Rossum show that neural networks can overcome this limitation by storing memories initially in a transient form that does not require protein synthesis. Doing so reduces energy requirements by as much as 10-fold.

Studies in living brains have shown that transient memories of this type do in fact exist. The current results hence offer a hypothesis as to how the brain can learn in a more energy efficient way. Energy consumption is thought to have placed constraints on brain evolution. It is also often a bottleneck in computers. By revealing how the brain encodes memories energy efficiently, the current findings could thus also inspire new engineering solutions.

metabolically highly inefficient. Therefore, we introduce a synaptic caching algorithm that is consistent with synaptic consolidation experiments, and distributes learning over transient and persistent synaptic changes. This algorithm increases efficiency manifold. Synaptic caching yields a novel interpretation to various aspects of synaptic physiology, and suggests more energy efficient neuromorphic designs.

## Results

### Inefficiency of perceptron learning

To examine the metabolic energy cost associated to synaptic plasticity, we first study the perceptron. A perceptron is a single artificial neuron that attempts to binary classify input patterns. It forms the core of many artificial networks and has been used to model plasticity in cerebellar Purkinje cells. We consider the common case where the input patterns are random patterns each associated to a randomly chosen binary output. Upon presentation of a pattern, the perceptron output is calculated and compared to the desired output. The synaptic weights are modified according to the perceptron learning rule, *Figure 1A*. This is repeated until all patterns are classified correctly (*Rosenblatt, 1962*, see Materials and methods). Typically, the learning takes multiple iterations over the whole dataset ('epochs').

As it is not well known how much metabolic energy is required to modify a biological synapse, and how this depends on the amount of change and the sign of the change, we propose a parsimonious model. We assume that the metabolic energy for every modification of a synaptic weight is proportional to the amount of change, no matter if this is positive or negative. The total metabolic cost $M$ (in arbitrary units) to train a perceptron is the sum over the weight changes of synapses

$$M_{\mathrm{perc}} = \sum_{i=1}^{N} \sum_{t=1}^{T} |w_i(t) - w_i(t-1)|^{\alpha}, \qquad (1)$$

where $N$ is the number of synapses, $w_i$ denotes the synaptic weight at synapse $i$, and $T$ is the total number of time-steps required to learn the classification. The exponent $\alpha$ is set to one, but our

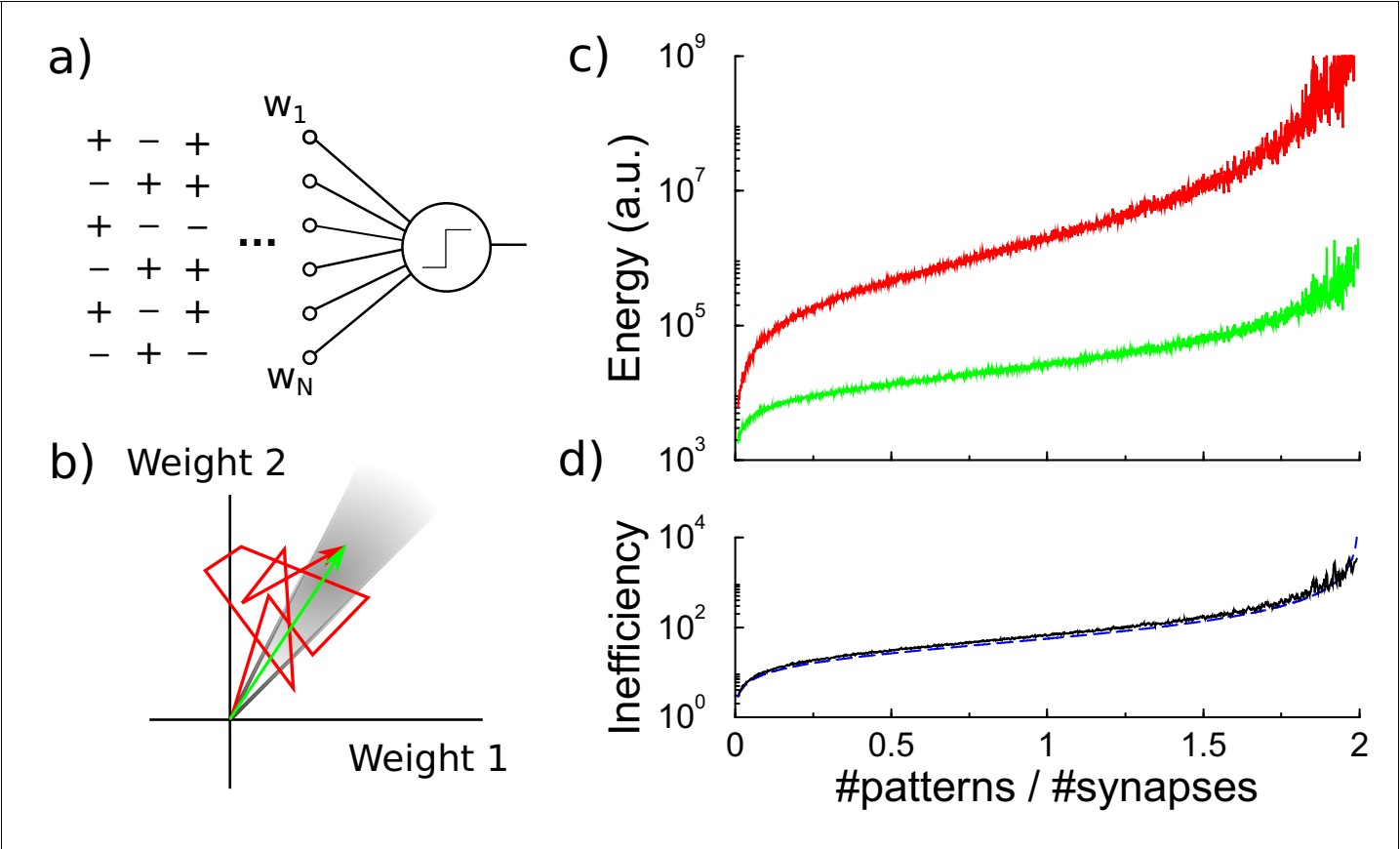

**Figure 1.** Energy efficiency of perceptron learning. (**a**) A perceptron cycles through the patterns and updates its synaptic weights until all patterns produce their correct target output. (**b**) During learning the synaptic weights follow approximately a random walk (red path) until they find the solution (grey region). The energy consumed by the learning corresponds to the total length of the path (under the $L_1$ norm). (**c**) The energy required to train the perceptron diverges when storing many patterns (red curve). The minimal energy required to reach the correct weight configuration is shown for comparison (green curve). (**d**) The inefficiency, defined as the ratio between actual and minimal energy plotted in panel c, diverges as well (black curve). The overlapping blue curve corresponds to the theory, *Equation 3* in the text.

The online version of this article includes the following figure supplement(s) for figure 1:

**Figure supplement 1.** Energy inefficiency as a function of exponent $\alpha$ in the energy function.

results below are similar whenever $0 \leq \alpha \leq 2$, *Figure 1—figure supplement 1*. As there is evidence that synaptic depression involves different pathways than synaptic potentiation (e.g. *Hafner et al., 2019*), we also tried a variant of the cost function where only potentiation costs energy and depression does not. This does not change our results, *Figure 1—figure supplement 1*.

Learning can be understood as a search in the space of synaptic weights for a weight vector that leads to correct classification of all patterns, *Figure 1B*. The synaptic weights approximately follow a random walk (Materials and methods), and the metabolic cost is proportional to the length of this walk under the $L_1$ norm, *Equation 1*. The perceptron learning rule is energy inefficient, because repeatedly, weight modifications made to correctly classify one pattern are partly undone when learning another pattern. However, as both processes require energy, this is inefficient.

The energy required by the perceptron learning rule depends on the number of patterns $P$ to be classified. The set of correct weights spans a cone in $N$-dimensional space (grey region in *Figure 1B*). As the number of patterns to be classified increases, the cone containing correct weights shrinks and the random walk becomes longer (*Gardner, 1987*). Near the critical capacity of the perceptron ($P = 2N$), the number of epochs required diverges as $(2 - P/N)^{-2}$, *Opper (1988)*. The energy required, which is proportional to the number of updates that the weights undergo, follows a similar behavior, *Figure 1C*.

It is useful to consider the theoretical minimal energy required to classify all patterns. The most energy efficient algorithm would somehow directly set the synaptic weights to their desired final values. Geometrically, the random walk trajectory of the synaptic weights to the target is replaced by a path straight to the correct weights (green arrow in *Figure 1B*). Given the initial weights $w_i(0)$ and the final weights $w_i(T)$, the energy required in this idealized case is

$$M_{\min} = \sum_i |w_i(T) - w_i(0)|. \tag{2}$$

While the minimal energy also grows with memory load (Materials and methods), it increases less steeply, *Figure 1C*.

We express the metabolic efficiency of a learning algorithm as the ratio between the energy the algorithm requires and the minimal energy (the gap between the two log-scale curves in *Figure 1C*). As the number of patterns increases, the inefficiency of the perceptron rule rapidly grows as (see Materials and methods)

$$\frac{M_{\mathrm{perc}}}{M_{\min}} = \frac{\sqrt{\pi P}}{2 - P/N}, \tag{3}$$

which fits the simulations very well, *Figure 1D*, black curve and dashed blue curve.

There is evidence that both cerebellar and cortical neurons are operating close to their maximal memory capacity (*Brunel et al., 2004*; *Brunel, 2016*). Indeed, it would appear wasteful if this were not the case. However, the above result demonstrates that for instance classifying 1900 patterns by a neuron with 1000 synapses with the traditional perceptron learning requires about ~900 times more energy than minimally required. As the fruit-fly experiments indicate that even storing a single association in long-term memory is already metabolically expensive, storing many memories would thus require very large amounts of energy if the biology would naively implement these learning rules.

## Synaptic caching

How can the conflicting demands of energy efficiency and high storage capacity be met? The minimal energy argument presented above suggests a way to increase energy efficiency. There are forms of plasticity – anesthesia-resistant memory in flies and early-LTP/LTD in mammals – that decay and do not require protein synthesis. Such transient synaptic changes can be induced using a massed, instead of a spaced, stimulus presentation protocol. Fruit-fly experiments show that this form of plasticity is much less energy-demanding than long-term memory (*Mery and Kawecki, 2005*; *Plaçais and Preat, 2013*; *Plaçais et al., 2017*). In mammals, there is evidence that synaptic consolidation, but not transient plasticity, is suppressed under low-energy conditions (*Potter et al., 2010*). Inspired by these findings, we propose that the transient form of plasticity constitutes a synaptic variable that accumulates the synaptic changes across multiple updates in a less expensive transient form of memory; only occasionally the changes are consolidated. We call this *synaptic caching*.

Specifically, we assume that each synapse is comprised of a transient component $s_i$ and a persistent component $l_i$. The total synaptic weight is their sum, $w_i = s_i + l_i$. We implement synaptic caching as follows, *Figure 2A*: For every presented pattern, changes in the synaptic strength are calculated according to the perceptron rule and are accumulated in the transient component that decays exponentially to zero. If, however, the absolute value of the transient component of a synapse exceeds a certain consolidation threshold, all synapses of that neuron are consolidated (vertical dashed line in *Figure 2A*); the value of the transient component is added to the persistent weight; and the transient weight is reset to zero.

The efficiency gain of synaptic caching depends on the limitations of transient plasticity. If the transient synaptic component could store information indefinitely at no metabolic cost, consolidation could be postponed until the end of learning and the energy would equal the minimal energy *Equation 2*. Hence the efficiency gain would be maximal. However, we assume that the efficiency gain of synaptic caching is limited because of two effects: (1) The transient component decays exponentially (with a time-constant τ). (2) There might be a maintenance cost associated to maintaining the transient component. Biophysically, transient plasticity might correspond to an increased/decreased

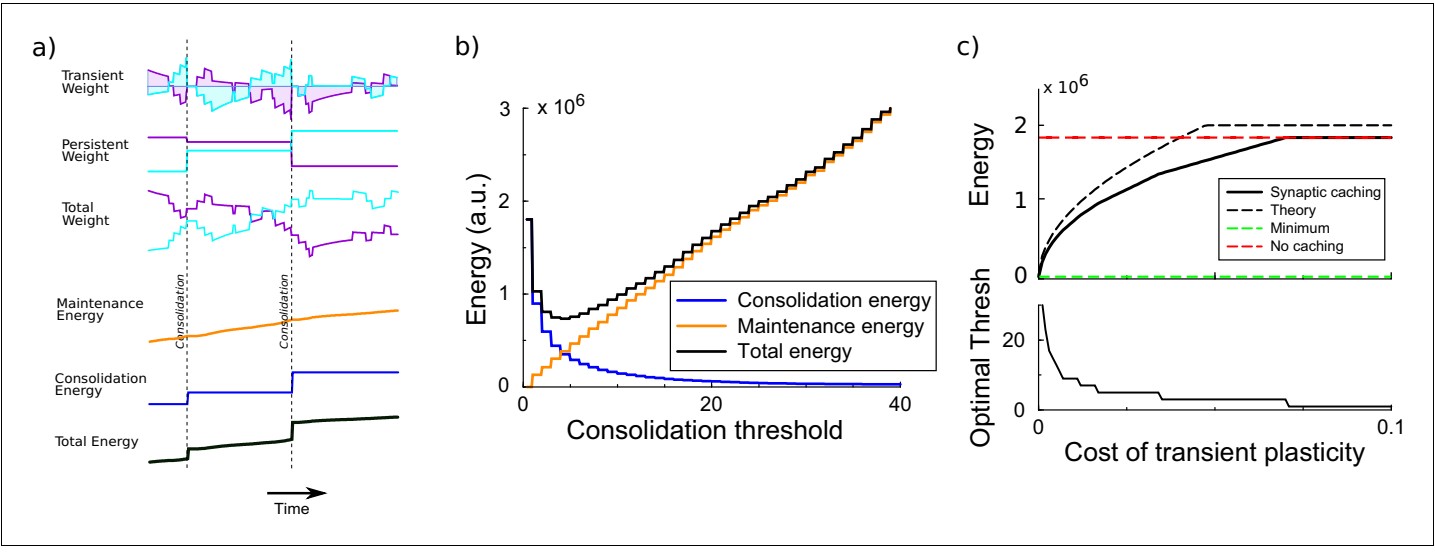

**Figure 2.** Synaptic caching algorithm. (**a**) Changes in the synaptic weights are initially stored in metabolically cheaper transient decaying weights. Here, two example weight traces are shown (blue and magenta). The total synaptic weight is composed of transient and persistent forms. Whenever any of the transient weights exceed the consolidation threshold, the weights become persistent and the transient values are reset (vertical dashed line). The corresponding energy consumed during the learning process consists of two terms: the energy cost of maintenance is assumed to be proportional to the magnitude of the transient weight (shaded area in top traces); energy cost for consolidation is incurred at consolidation events. (**b**) The total energy is composed of the energy to occasionally consolidate and the energy to support transient plasticity. Here, it is minimal for an intermediate consolidation threshold. (**c**) The amount of energy required for learning with synaptic caching, in the absence of decay of the transient weights (black curve). When there is no decay and no maintenance cost, the energy equals the minimal one (green line) and the efficiency gain is maximal. As the maintenance cost increases, the optimal consolidation threshold decreases (lower panel) and the total energy required increases, until no efficiency is gained at all by synaptic caching.

The online version of this article includes the following figure supplement(s) for figure 2:

**Figure supplement 1.** Synaptic caching in a spiking neuron with a biologically plausible perceptron-like learning rule.

vesicle release rate (*Padamsey and Emptage, 2014*; *Costa et al., 2015*) so that it diverges from its optimal value (*Levy and Baxter, 2002*).

To estimate the energy saved by synaptic caching, we assume that the maintenance cost is proportional to the transient weight itself and incurred every time-step $\Delta t$ (shaded area in the top traces of *Figure 2A*)

$$M_{\mathrm{trans}} = c \sum_i \sum_t |s_i(t)|.$$

While experiments indicate that transient plasticity is metabolically far less demanding than the persistent form, the precise value of the maintenance cost is not known. We encode it in the constant $c$; the theory also includes the case that $c$ is zero. It is straightforward to include a cost term for changing the transient weight (Materials and methods); such a cost would reduce the efficiency gain attainable by synaptic caching.

Next, we need to include the energetic cost of consolidation. Currently it is unknown how different components of synaptic consolidation, such as signaling, protein synthesis, transport to the synapses and changing the synapse, contribute to this cost. We assume the metabolic cost to consolidate the synaptic weights is $M_{\mathrm{cons}} = \sum_i \sum_t |l_i(t) - l_i(t-1)|$. This is identical to *Equation 1*, but in contrast to standard perceptron learning where synapses are consolidated every time a weight is updated, now changes in the persistent component $l_i$ only occur when consolidation occurs. One could add a maintenance cost term to the persistent weight as well, in that case postponing consolidation would save even more energy.

## Efficiency gain from synaptic caching

To maximize the efficiency gain achieved by synaptic caching one needs to tune the consolidation threshold, *Figure 2B*. When the threshold is low, consolidation occurs often and the energy approaches the one without synaptic caching. When on the other hand the consolidation threshold is high, the expensive consolidation process occurs rarely, but the maintenance cost of transient plasticity is high; moreover, the decay will lead to forgetting of unconsolidated memories, slowing down learning and increasing the energy cost. Thus, the consolidation energy decreases for larger thresholds, whereas the maintenance energy increases, *Figure 2B* (see Materials and methods). As a result of this trade-off, there is an optimal threshold – which depends on the decay and the maintenance cost – that balances persistent and transient forms of plasticity. To analyze the efficiency gain below, we numerically optimize the consolidation threshold.

First, we consider the case when the transient component does not decay. *Figure 2C* shows the energy required to train the perceptron. When the maintenance cost is absent ($c = 0$), consolidation is best postponed until the end of the learning and the energy is as low as the theoretical minimal bound. As $c$ increases, it becomes beneficial to consolidate more often, that is the optimal threshold decreases, *Figure 2C* bottom panel. The required energy increases until the maintenance cost becomes so high that it is better to consolidate after every update, the transient weights are not used, and no energy is saved with synaptic caching. The efficiency is well estimated by analysis presented in the Materials and methods, *Figure 2C* (theory).

Next, we consider what happens when the transient plasticity decays. We examine the energy and learning time as a function of the decay rate for various levels of maintenance cost, *Figure 3*. As stated above, if there is no decay, efficiency gain can be very high; the consolidation threshold has no impact on the learning time, *Figure 3* bottom. In the other limit, when the decay is rapid (rightmost region), it is best to consolidate frequently as otherwise information is lost. As expected, the metabolic cost is high in this case.

The regime of intermediate decay is quite interesting. When maintenance cost is high, it is of primary importance to keep learning time short, and in fact the learning time can be lower than in a perceptron without decay, *Figure 3*, bottom, light curves. When on the other hand maintenance cost is low, the optimal solution is to set the consolidation threshold high so as to minimize the number of consolidation events, even if this means a longer learning time, *Figure 3*, bottom, dark curves.

For intermediate decay rates, the consolidation threshold trades off between learning time and energy efficiency, *Figure 3—figure supplement 1A*. That is, by setting the consolidation threshold the perceptron can learn either rapidly or efficiently. Such a trade-off could be of biological relevance. We found a similar trade-off in multi-layer perceptrons (see below), *Figure 3—figure supplement 1B*. (although we found no evidence that learning can be sped up there).

In summary, when the transient component decays the learning dynamics is altered, and synaptic caching can not only reduce metabolic cost but can also reduce learning time.

Next, to show that synaptic caching is a general principle, we implement synaptic caching in a spiking neural network with a biologically plausible perceptron-like learning rule proposed by *D'Souza et al. (2010)*. The optimal scenario, where the transient weights do not decay and have no maintenance cost, is assumed. The network is able to save 80% of the energy with synaptic caching, *Figure 2—figure supplement 1*. Hence, efficiency gains from synaptic caching do not rely on exact implementation.

In the above implementation of synaptic caching, consolidation of all synapses was triggered when transient plasticity at a single synapse exceeded a certain threshold. This resembles the synaptic tagging and capture phenomenon where plasticity induction leads to transient changes and sets a tag; only strong enough stimulation results in proteins being synthesized and being delivered to all tagged synapses, consolidating the changes (*Frey and Morris, 1997*; *Barrett et al., 2009*). There is a number of ways synapses could interact, *Figure 4A*. First, consolidation might be set to occur whenever transient plasticity at a synapse crosses the threshold and only that synapse is consolidated. Second, a hypothetical signal might send to the soma and consolidation of all synapses occurs once transient plasticity at any synapse crosses the threshold (used in *Figures 2* and *5*). Third, a hypothetical signal might be accumulated in or near the soma and consolidation of all synapses occurs once this total transient plasticity across synapses crosses the threshold. Only cases 2 and 3

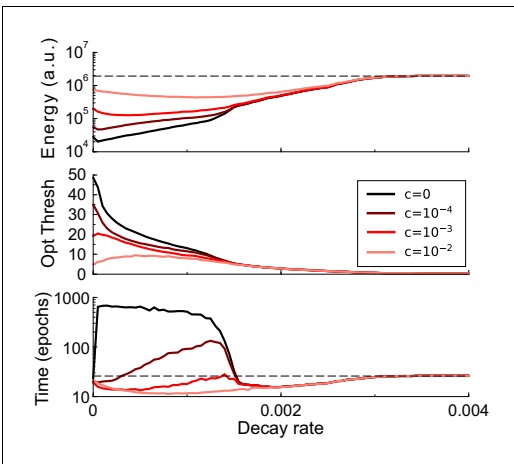

**Figure 3.** Synaptic caching and decaying transient plasticity. The amount of energy required, the optimal consolidation threshold, and the learning time as a function of the decay rate of transient plasticity for various values of the maintenance cost. Broadly, stronger decay will increase the energy required and hence reduce efficiency. With weak decay and small maintenance cost, the most energy-saving strategy is to accumulate as many changes in the transient forms as possible, thus increasing the learning time (darker curves). However, when maintenance cost is high, it is optimal to reduce the threshold and hence learning time. Dashed lines denote the results without synaptic caching.

The online version of this article includes the following figure supplement(s) for figure 3:

**Figure supplement 1.** The effects of consolidation threshold on energy cost and learning time.

are consistent with synaptic tagging and capture experiments, where consolidation of one synapse also leads to consolidation of another synapse that would otherwise decay back to baseline (**Frey and Morris, 1997**; **Sajikumar et al., 2005**). However, all variants lead to comparable efficiency gains, **Figure 4B**.

In summary, we see that synaptic caching can in principle achieve large efficiency gains, bringing efficiency close to the theoretical minimum.

## Synaptic caching in multilayer networks

Since the perceptron is a rather restrictive framework, we wondered whether the efficiency gain of synaptic caching can be transferred to multilayer networks. Therefore, we implement a multilayer network trained with back-propagation. Back-propagation networks learn the associations of patterns by approaching the minimum of the error function through stochastic gradient descent. We use a network with one hidden layer with by default 100 units to classify hand-written digits from the MNIST dataset. As we train the network, we intermittently interrupt the learning to measure the energy consumed for plasticity thus far and measure the performance on a held-out test-set. This yields a curve relating energy to accuracy.

Similar to a perceptron, learning without synaptic caching is metabolically expensive in a back-propagation network. Until reaching maximal accuracy, energy rises approximately exponentially with accuracy, after which additional energy do not lead to further improvement. When the learning rate is sufficiently small, the metabolic cost of plasticity is independent of the learning rate. At larger learning rates, learning no longer converges and energy goes up steeply without an increase in accuracy, **Figure 5A**. With the exception of these very large rates, these results show that lowering the learning rate does not save energy.

Similar to the perceptron, we evaluate how much energy would be required to directly set the synaptic weights to their final values. Traditional learning without synaptic caching is once again energetically inefficient, expending at least ~20 times more energy compared to this theoretical minimum whatever the desired accuracy level is, **Figure 5B**. However, by splitting the weights into persistent synaptic weights and transient synaptic caching weights, the network can save substantial amounts of energy. As for the perceptron, depending on the decay and the maintenance cost the energy ranges from as little as the minimum to as much as the energy required without caching. Thus, the efficiency gain of synaptic caching found for the perceptron carries over to multilayer networks.

It might seem that smaller networks would be metabolically less costly, because small networks simply contain fewer synapses to modify. On the other hand, we saw above that for the perceptron metabolic costs rise rapidly when cramming many patterns into it. We wondered therefore how energy cost depends on network size in the multilayer network. Since the number of input units is fixed to the image size and the number of output units equals the ten output categories, we adjust the number of hidden units.

The network fails to reach the desired accuracy if the number of hidden units is too small, **Figure 5C**. When the network size is barely above the minimum requirement, the network has to

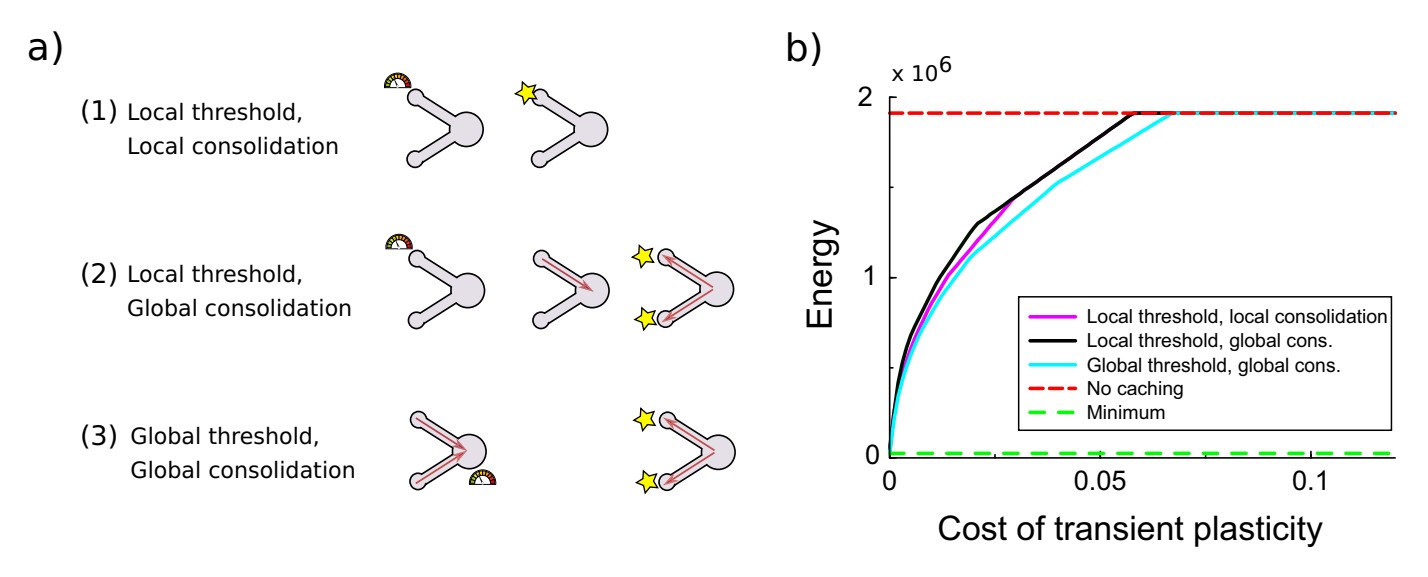

**Figure 4.** Comparison of various variants of the synaptic caching algorithm. (a) Schematic representation of variants to decide when consolidation occurs. From top to bottom: (1) Consolidation (indicated by the star) occurs whenever transient plasticity at a synapse crosses the consolidation threshold and only that synapse is consolidated. (2) Consolidation of all synapses occurs once transient plasticity at any synapse crosses the threshold. (3) Consolidation of all synapses occurs once the total transient plasticity across synapses crosses the threshold. (b) Energy required to teach the perceptron is comparable across algorithm variants. Consolidation thresholds were optimized for each algorithm and each maintenance cost of transient plasticity individually. In this simulation the transient plasticity did not decay.

compensate the lack of hidden units with longer training time and hence a larger energy expenditure. However, very large networks also require more energy. These results show that from an energy perspective there exists an optimal number of neurons to participate in memory formation. The optimal number depends on the accuracy requirement; as expected, higher accuracies require more hidden units and energy.

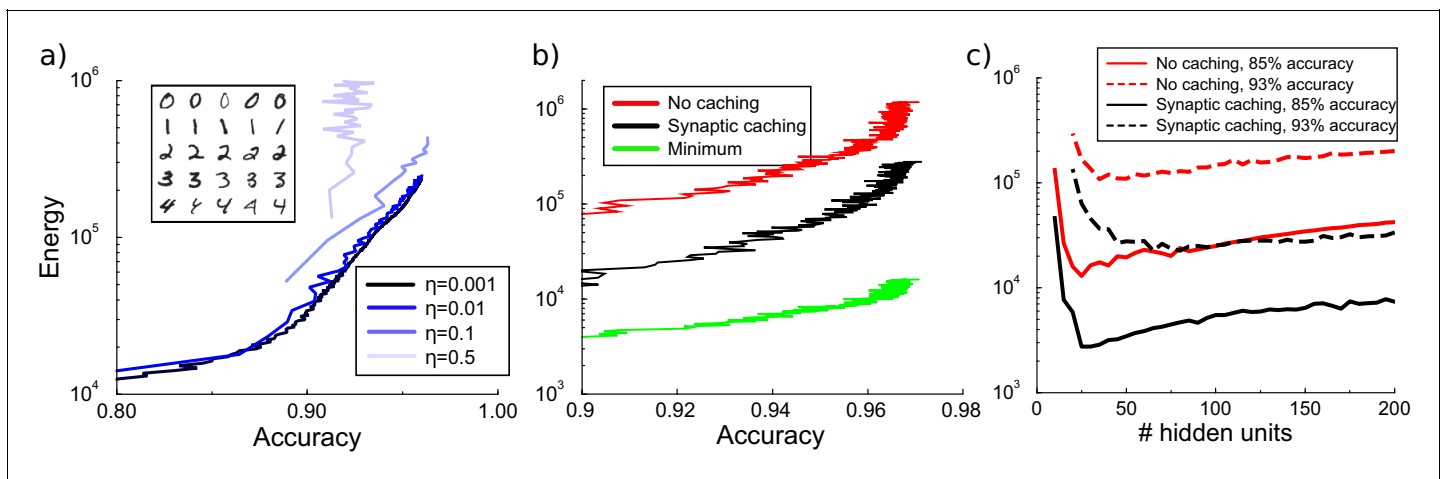

**Figure 5.** Energy cost to train a multilayer back-propagation network to classify digits from the MNIST data set. (a) Energy rises with the accuracy of identifying the digits from a held-out test data. Except for the larger learning rates, the energy is independent of the learning rate $\eta$. Inset shows some MNIST examples. (b) Comparison of energy required to train the network with/without synaptic caching, and the minimal energy. As for the perceptron and depending on the cost of transient plasticity, synaptic caching can reduce energy need manifold. (c) There is an optimal number of hidden units that minimizes metabolic cost. Both with and without synaptic caching, energy needs are high when the number of hidden units is barely sufficient or very large. Parameters for transient plasticity in (b) and (c): $\eta = 0.1$, $\tau = 1000$, $c = 0.001$.

## Discussion

Experiments on formation of a long-term memory of a single association suggest that synaptic plasticity is an energetically expensive process. We have shown that energy requirements rise steeply as memory load or designated accuracy level increase. This indicates trade-offs between energy consumption, and network capacity and performance. To improve efficiency, we have proposed an algorithm named synaptic caching that temporarily stores changes in the synaptic strength in transient forms of plasticity, and only occasionally consolidates into the persistent forms. Depending on the characteristics (decay and maintenance cost) of transient plasticity, this can lead to large energy savings in the energy required for synaptic plasticity. We stress that from an algorithmic point of view, synaptic caching can be applied to any synaptic learning algorithm (unsupervised, reinforcement, supervised) and does not have specific requirements. Further savings might be possible by adjusting the consolidation threshold as learning progresses and by being pathway-specific (*Leibold and Monsalve-Mercado, 2016*).

The implementation of a consolidation threshold is similar to what has been observed in physiology, in particular in the synaptic tagging and capture literature (*Redondo and Morris, 2011*). Our results thus give a novel interpretation of those findings. Synaptic consolidation is known to be affected by reward, novelty and punishment (*Redondo and Morris, 2011*), which is compatible with a metabolic perspective as energy is expended only when the stimulus is worth remembering. In addition, our results for instance explain why consolidation is competitive, but transient plasticity is less so (*Sajikumar et al., 2014*), namely the formation of long-term memory is precious. Consistent with this, there is evidence that encouraging consolidation increases energy consumption (*Plaçais et al., 2017*). We also predict that the transient weight changes act as an accumulative threshold for consolidation. That is, sufficient transient plasticity should trigger consolidation, even in the absence of other consolidation triggers. Future characterization of the energy budget of synaptic plasticity should allow more precise predictions of our theory.

Combining persistent and transient storage mechanisms is a strategy well known in traditional computer systems to provide a faster and often energetically cheaper access to memory. In computer systems, permanent storage of memories typically requires transmission of all information across multiple transient cache systems until reaching a long-term storage device. The transfer of information is often a bottleneck in computer architectures and consumes considerable power in modern computers (*Kestor et al., 2013*). However, in the nervous system transient and persistent synapses appear to exist next to each other. Local consolidation in a synapse does not require moving information. Using this setup, biology appears to have found a more efficient way to store information.

Memory stability has long fascinated researchers (*Richards and Frankland, 2017*), and in some cases forgetting can be beneficial (*Brea et al., 2014*). Splitting plasticity into transient and persistent forms might prevent catastrophic forgetting in networks (*Leimer et al., 2019*). Here, we argue that the main benefit of more transient forms of plasticity is to permit the network to explore the weight space to find a desirable weight configuration using less energy. While this work focuses solely on the metabolic cost of synaptic plasticity, the brain also expends significant amounts of energy on spiking, synaptic transmission, and maintaining resting potential. Learning rules can be designed to reduce costs associated to computation once learning has finished (*Sacramento et al., 2015*). It would be of interest to next understand the precise interaction of computation and plasticity cost during and after learning.

## Materials and methods

### Energy efficiency of the perceptron

For perceptron, we can calculate the energy efficiency of both the classical perceptron and the gain achieved by synaptic caching. We first consider the case that transient plasticity does not decay, as this allows important theoretical simplifications. In the perceptron learning to classify binary patterns *Equation 8*, the weight updates are either $+\eta$ or $-\eta$, where $\eta$ is the learning rate, so that the energy spent (*Equation 1*, $\alpha = 1$) per update per synapse equals $\eta$. Hence the total energy spent to classify all patterns $M_{\mathrm{perc}} = NK\eta$, where $K$ is the total number of updates. *Opper (1988)* showed that

learning time diverges as $K \sim (2 - P/N)^{-2}$. We found the numerator numerically to yield $K = 2P/(2 - P/N)^2$.

To calculate the efficiency, we compare this to the minimal energy necessary to reach the final weight vector in the perceptron. We approximate the weight trajectory followed by the perceptron algorithm by a random walk. After $K$ updates of step-size $\eta$ the weights approximate a Gaussian distribution with zero mean and variance $K\eta^2$. By short-cutting the random walk, the minimal energy required to reach the weight vector is $M_{\min} = N\langle|w_i|\rangle = \sqrt{\frac{2}{\pi}}\eta N\sqrt{K}$. Hence, we find for the inefficiency (see *Figure 1D*)

$$\frac{M_{\mathrm{perc}}}{M_{\min}} = \frac{\sqrt{\pi P}}{2 - P/N}.$$

Simulations show that the variance in the weights is actually about 20% smaller than a random walk, likely reflecting correlations in the learning process not captured in the random walk approximation. This explains most of the slight deviation in the inefficiency between theory and simulation, *Figure 1D*.

## Efficiency of synaptic caching

To calculate the efficiency gained with synaptic caching, we need to calculate both the consolidation energy and the maintenance energy. The consolidation energy equals the number of consolidation events times the size of the updates. The size of the weight updates is equal to the consolidation threshold $\theta$, while the number of consolidation events follows from a random walk argument as $NK/\lceil\theta/\eta\rceil^2$. The ceiling function expresses the fact that when the threshold is smaller than learning rate, consolidation will always occur; we temporarily ignore this scenario. In addition, at the end of learning all remaining transient plasticity is consolidated, which requires an energy $N\langle|s_i(T)|\rangle$. Assuming that the probability distribution of transient weights, $P_s(s)$, has reached steady state at the end of learning, it has a triangular shape (see below) and mean absolute value $\langle|s_i(T)|\rangle = \frac{1}{3}\theta$, so that the total consolidation energy

$$M_{\mathrm{cons}} = \eta^2 \frac{NK}{\theta} + \frac{1}{3}N\theta.$$

The energy associated to the transient plasticity is (again assuming that $P_s(s)$ has reached steady state)

$$M_{\mathrm{trans}} = cNT\theta/3, \qquad (4)$$

where $T$ is the number of time-steps required for learning. We find numerically that $T = \frac{P^{3/2}}{(2-P/N)^2}$. Hence the total energy when using synaptic caching is $M_{\mathrm{cache}} = M_{\mathrm{cons}} + M_{\mathrm{trans}} = N[\eta^2 K/\theta + \frac{1}{3}\theta(1 + cT)]$. The optimal threshold $\hat{\theta}$ is given by $\frac{d}{d\theta}[M_{\mathrm{cons}} + M_{\mathrm{trans}}] = 0$, or

$$\hat{\theta}^2 = \eta^2 \frac{3K}{1 + cT}$$

at which the energy is $M_{\mathrm{cache}} = 2\eta N\sqrt{K}\sqrt{1 + cT}/\sqrt{3}$. And so the efficiency of synaptic caching is $\frac{M_{\mathrm{cache}}}{M_{\min}} = \sqrt{\frac{2\pi}{3}}\sqrt{1 + cT}$. However, as consolidation can maximally occur only once per time-step, $M_{\mathrm{cons}}$ cannot exceed $M_{\mathrm{perc}}$ so that the inefficiency is

$$\frac{M_{\mathrm{cache}}}{M_{\min}} = \min\left(\sqrt{\frac{2\pi}{3}(1 + cT)}, \sqrt{\frac{\pi}{2}K}\right).$$

This equation reasonably matches the simulations, *Figure 2C* (labeled 'theory').

One can include a cost for changing the transient weight, so that $M_{\mathrm{trans}} = c\sum_i \sum_t |s_i(t)| + b\sum_i \sum_t |s_i(t+1) - s_i(t)|$, where $b$ codes the cost of making a change. Assuming that consolidating immediately after a weight change does not incur this cost, this yields an extra

term in *Equation4* of $bNK(1 - 1/\lceil\theta/\eta\rceil^2)$. Such costs will reduce the efficiency gain achievable by synaptic caching. When $b \geq 1$, it is always cheaper to consolidate.

## Decaying transient plasticity

When transient plasticity decays, the situation is more complicated as the learning time depends on the strength of the decay and to our knowledge no analytical expression exists for it. However, it is still possible to estimate the *power*, that is the energy per time unit, for both the transient component, denoted $m_{\text{trans}}$, and the consolidation component, $m_{\text{cons}}$. Under the random walk approximation every time the perceptron output does not match the desired output, the transient weight $s_i$ is updated with an amount $\Delta s_i$ drawn from a distribution $Q$, with zero mean and variance $\sigma^2$. Given the update probability $p$, that is the fraction of patterns not yet classified correctly, one has $Q_s(\eta) = Q_s(-\eta) = p/2$ and $Q_s(0) = 1 - p$, so that $\sigma_s^2 = p\eta^2$. We assume that the synaptic update rate decreases very slowly as learning progresses, hence $p$ is quasi-stationary.

Every time-step $\Delta t = 1$ the transient weights decay with a time-constant $\tau$. The synapse is consolidated and $s_i$ is reset to zero whenever the absolute value of the caching weight $|s_i|$ exceeds $\theta$. Given $p$ and $\tau$, we would like to know: 1) how often consolidation events occur which gives consolidation power and 2) the maintenance power $m_{\text{trans}} = cN\langle|s_i|\rangle$. This problem is similar to the random walk to threshold model used for integrate-and-fire neurons, but here there are two thresholds: $\theta$ and $-\theta$.

Under the assumptions of small updates and a smooth resulting distribution, the evolution of the probability distribution $P_s(s_i)$ is described by the Fokker-Planck equation, which in the steady state gives

$$0 = -\frac{1}{\tau}\frac{\partial}{\partial s_i}[s_i P_s(s_i)] + \frac{1}{2}\sigma_s^2\frac{\partial^2}{\partial s_i^2}P_s(s_i) + r\delta(s_i).$$

The last term is a source term that describes the re-insertion of weights by the reset process. The boundary conditions are $P_s(s_i = \pm\theta) = 0$. While $P_s(s_i)$ is continuous in $s_i$, the source introduces a cusp in $P_s(s_i)$ at the reset value. Conservation of probability ensures that $r$ equals the outgoing flux at the boundaries. One finds

$$P_s(s_i) = \frac{1}{Z}\exp\left[-\frac{s_i^2}{\sigma^2}\right]\left[\text{erfi}\left(\frac{|s_i|}{\sigma}\right) - \text{erfi}\left(\frac{\theta}{\sigma}\right)\right],$$

where $\text{erfi}(x) = -i\text{erf}(ix)$, $\sigma^2 = \frac{\tau}{\Delta t}\sigma_s^2$ and with normalization factor

$$Z = \frac{2\theta^2}{\sqrt{\pi}\sigma}{}_2F_2\left(1,1;\frac{3}{2},2;-(\frac{\theta}{\sigma})^2\right) - \sqrt{\pi}\sigma\text{erf}\left(\frac{\theta}{\sigma}\right)\text{erfi}\left(\frac{\theta}{\sigma}\right),$$

where ${}_2F_2$ is the generalized hypergeometric function. (In the limit of no decay this becomes a triangular distribution $P_s(s_i) = [\theta - |s_i|]/\theta^2$.)

We obtain maintenance power

$$m_{\text{trans}} = cN\langle|s_i|\rangle \tag{5}$$

$$= \frac{cN}{Z}\left[\frac{2\theta\sigma}{\sqrt{\pi}} - \sigma^2\text{erfi}\left(\frac{\theta}{\sigma}\right)\right]. \tag{6}$$

For small $\theta/\sigma$, that is small decay, this is linear in $\theta$, $m_{\text{trans}} \approx \frac{cN\theta}{3}$. It saturates for large $\theta$ because then the decay dominates and the threshold is hardly ever reached.

The consolidation rate follows from Fick's law

$$r = \frac{1}{2}\sigma^2 P'_s(-\theta) - \frac{1}{2}\sigma^2 P'_s(\theta)$$
$$= \frac{-2\sigma}{Z\sqrt{\pi}}.$$

The consolidation power is

$$m_{\text{cons}} = N\theta r. \tag{7}$$

In the limit of no decay one has $r = \sigma^2/\theta^2$, so that $m_{\text{cons}} = pN\eta^2/\theta$. Strictly speaking this approximates learning with a random walk process and assumes local consolidation, *Figure 4A*. However, *Equations 6 and 7* give a good prediction of the simulation when provided with the time-varying update probability from the simulation, *Figure 6*.

## Simulations

### Perceptron

Unless stated otherwise, we use a perceptron with $N = 1000$ input units to classify $P = N$ random binary ($\pm 1$ with equal probability) input patterns $x^{(p)}$, each to be associated to a randomly assigned desired binary output $d^{(p)}$. Each input unit is connected with a weight $w_i$ signifying the strength of the connection. An 'always-on' bias unit with corresponding weight is included to adjust the threshold of the perceptron. The perceptron output $y$ of a pattern is determined by the Heaviside step function $\Theta$, $y = \Theta(\text{w}.x)$. If for a given pattern $p$, the output does not match the desired pattern output, $w$ is adjusted according to

$$\Delta w_i = \eta\left(d^{(p)} - y^{(p)}\right)x_i^{(p)}, \qquad (8)$$

where the learning rate $\eta$ can be set to one without loss of generality. The perceptron algorithm cycles through all patterns until classified correctly. In principle, the magnitude of the weight vector, and hence the minimal energy, can be arbitrarily small for a noise-free binary perceptron. However, this paradox is resolved as soon as robustness to any post-synaptic noise is required.

### Multilayer networks

For the multilayer networks trained on MNIST, we use networks with one hidden layer, logistic units, and one-hot encoding at the output. Weights are updated according to the mean squared error back-propagation rule without regularization.

Simulation scripts for both the perceptron and the multilayer network can be found at https://github.com/vanrossumlab/li_vanrossum_19. (*Li and van Rossum, 2020*; copy archived at https://github.com/elifesciences-publications/li_vanrossum_19).

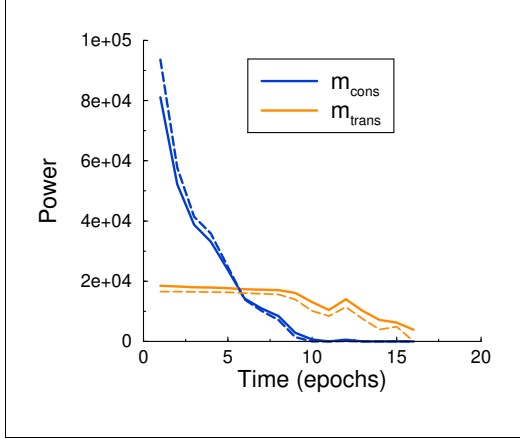

**Figure 6.** Maintenance and consolidation power. Power (energy per epoch) of the perceptron vs epoch. Solid curves are from simulation, dashed curves are the theoretical predictions, *Equations 6 and 7*, with $\sigma$ calculated by using the perceptron update rate $p$ extracted from the simulation. Both powers are well described by the theory. Parameters: $\tau = 500$, $c = 0.01$, $\theta = 5$.

## Acknowledgements

This project is supported by the Leverhulme Trust with grant number RPG-2017–404. MvR is supported by Engineering and Physical Sciences Research Council (EPSRC) grant EP/R030952/1. We would like to thank Joao Sacramento and Simon Laughlin for discussion and inputs.

## Additional information

### Competing interests

Mark CW van Rossum: Reviewing editor, *eLife*. The other author declares that no competing interests exist.

## Funding

| Funder | Grant reference number | Author |
|---|---|---|
| Leverhulme Trust | RPG-2017-404 | Mark CW van Rossum |
| Engineering and Physical Sciences Research Council | EP/R030952/1 | Mark CW van Rossum |

The funders had no role in study design, data collection and interpretation, or the decision to submit the work for publication.

## Author contributions

Ho Ling Li, Software, Investigation; Mark CW van Rossum, Conceptualization, Formal analysis, Supervision, Investigation

## Author ORCIDs

Ho Ling Li (ID) https://orcid.org/0000-0002-5654-0183
Mark CW van Rossum (ID) https://orcid.org/0000-0001-6525-6814

## Decision letter and Author response

Decision letter https://doi.org/10.7554/eLife.50804.sa1
Author response https://doi.org/10.7554/eLife.50804.sa2

## Additional files

### Supplementary files

• Transparent reporting form

### Data availability

Simulation scripts can be found at https://github.com/vanrossumlab/li_vanrossum_19 (copy archived at https://github.com/elifesciences-publications/li_vanrossum_19).

The following previously published dataset was used:

| Author(s) | Year | Dataset title | Dataset URL | Database and Identifier |
|---|---|---|---|---|
| LeCun Y, Cortes C, Burges CJC | 1999 | Data from: The MNIST database of handwritten digits | http://yann.lecun.com/exdb/mnist/ | The MNIST database of handwritten digits, mnist |

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
