## [Decision Letter]

**Acceptance summary:**

The authors investigate the issue of energy consumption in synapses. They show that under a strategy in which much of the information about weights is kept in temporary storage while permanent weight changes are rare, energy consumption can be reduced by an order of magnitude. There has been a great deal of work on energy consumption associated with action potentials and synaptic plasticity, but this is, to our knowledge, the first to consider energy efficiency in the context of learning. As such it fills an important gap in our understanding of synaptic plasticity. This paper should appeal to anybody who is interested either in synaptic plasticity or energy efficiency in the brain. It may also be important for learning in artificial systems, where energy costs for training networks that are small by brain standards can exceed millions of kilowatt-hours.

**Decision letter after peer review:**

Thank you for submitting your article "Energy efficient synaptic plasticity" for consideration by *eLife*. Your article has been reviewed by three peer reviewers, one of whom is a member of our Board of Reviewing Editors, and the evaluation has been overseen by Ronald Calabrese as the Senior Editor. The following individual involved in review of your submission has agreed to reveal their identity: Walter Senn (Reviewer #2).

The reviewers have discussed the reviews with one another and the Reviewing Editor has drafted this decision to help you prepare a revised submission.

Summary:

The authors investigate the issue of energy consumption in synapses. They show that under a strategy in which much of the information about weights is kept in temporary storage while permanent weight changes are rare, energy consumption can be reduced by an order of magnitude.

This is an interesting paper that formalizes the concept of energy efficiency in learning. While work exists that considers the energy consumption of action potentials and synaptic plasticity, the formalization of energy efficiency in the context of learning is new, and worth a publication. The paper is well written and the math seems to be sound.

Essential revisions:

1) The authors introduce two components of the synaptic strength, a quickly decaying component that is transcribed into a long-lasting component when a threshold is crossed. The quickly decaying component requires less energy, and thus learning becomes a trade-off between energy efficient storage in the decaying component and energy costly storage in the sustained component.

The decay will lead to forgetting of unconsolidated memories and to a slow-down of learning, together with an increase of energy. As far as we could tell, the paper does not consider the speed of learning, and instead only asks for an energy efficient learning up to a certain degree of accuracy. The authors should provide learning curves for different consolidation thresholds and decay rates. Ideal would be a plot of energy versus learning time – presumably the lower the energy, the longer it takes to achieve a given set of accuracy, although we admit that's only a guess. On the other hand, there may be an optimal threshold.

2) Previous work [e.g., Ziegler, Zenke,.…, Gerstner, 2015, "From synapses to behavioural modelling"; Zenke et al., 2017, "Continual Learning Through Synaptic Intelligence"] has shown a benefit for the 2-stage synapses model in terms of learning and forgetting. Is there a similar benefit for this 2-stage model? This may simply be a Discussion point, referring to the analysis asked in the point (1) above.

3) From a biological point of view, it is clear that the change as well as the maintenance of synaptic weight can cost energy. Nevertheless, we find it strange that the authors analyze the perceptron learning rule according to a change-only energy cost function while the synaptic caching rule is analyzed by a combination of change (late-phase) and maintenance (early-phase) cost function. How critical is the phase-cost relation? Is it also efficient if the late-phase costs maintenance (e.g., by having a larger synaptic apparatus) and the early-phase costs energy dependent on its change? What is the energy consumption of the perceptron learning rule considering the maintenance cost function?

4) Please test for comparison the energy consumption of other learning rules performing perceptron learning (e.g., D’Souza et al., 2010).

---

## [Author Response]

Essential revisions:1) The authors introduce two components of the synaptic strength, a quickly decaying component that is transcribed into a long-lasting component when a threshold is crossed. The quickly decaying component requires less energy, and thus learning becomes a trade-off between energy efficient storage in the decaying component and energy costly storage in the sustained component.The decay will lead to forgetting of unconsolidated memories and to a slow-down of learning, together with an increase of energy. As far as we could tell, the paper does not consider the speed of learning, and instead only asks for an energy efficient learning up to a certain degree of accuracy. The authors should provide learning curves for different consolidation thresholds and decay rates. Ideal would be a plot of energy versus learning time – presumably the lower the energy, the longer it takes to achieve a given set of accuracy, although we admit that's only a guess. On the other hand, there may be an optimal threshold.

The reviewers raise an interesting, subtle issue that we now discuss in much more detail in Results subsection “Efficiency gain from synaptic caching”. First, we first study a potential trade-off between energy and learning time in the perceptron. We now plot the relation between the learning time and the energy spend (Figure 3—figure supplement 1A). Depending on maintenance cost *c*, there are regimes where these form a trade-off:

When the decay is slow and *c* is small, the most energy-saving scenario is to accumulate as many changes as possible in the transient plasticity that still allow the perceptron to converge, resulting in long learning time. In this case there is a trade-off between learning time and energy.

However, when *c* is bigger, the perceptron has to choose an smaller threshold so that maintenance cost is limited. In this regime the energy-optimal threshold is close to the threshold that gives minimal learning time (Figure 3—figure supplement 1A).

Finally, when there is no decay, the energy cost depends on consolidation threshold, but the learning time does not change regardless of the threshold. Hence, there is no interaction between learning time and energy cost.

We also researched this issue in MLPs and found a similar picture: in some regimes there is a trade-off, but not in others (Figure 3—figure supplement 1B).

2) Previous work [e.g., Ziegler, Zenke,.…, Gerstner, 2015, "From synapses to behavioural modelling"; Zenke et al., 2017, "Continual Learning Through Synaptic Intelligence"] has shown a benefit for the 2-stage synapses model in terms of learning and forgetting. Is there a similar benefit for this 2-stage model? This may simply be a Discussion point, referring to the analysis asked in the point (1) above.

The main benefit of having 2-stage synapses is to save energy that would otherwise be expended if the network has to consolidate every single weight change. As described above, a decaying 2-state synapse model can under some circumstances reduce perceptron training time. However, we are not sure whether this generalizes to multi-layer perceptrons.

We now cite a recent study by Leimer, Herzog and Senn that shows how a similar 2-state model can prevent catastrophic forgetting (we thought this was the most relevant paper as the Zenke study does not employ a 2-state model, while Ziegler et al. do not show a functional advantage).

3) From a biological point of view, it is clear that the change as well as the maintenance of synaptic weight can cost energy. Nevertheless, we find it strange that the authors analyze the perceptron learning rule according to a change-only energy cost function while the synaptic caching rule is analyzed by a combination of change (late-phase) and maintenance (early-phase) cost function. How critical is the phase-cost relation? Is it also efficient if the late-phase costs maintenance (e.g., by having a larger synaptic apparatus) and the early-phase costs energy dependent on its change? What is the energy consumption of the perceptron learning rule considering the maintenance cost function?

We now discuss these two variants in detail:

If the transient component (early-phase) costs energy dependent on its change, the theory can be straightforwardly extended (see Materials and methods, subsection “Efficiency of synaptic caching”). As expected this extra cost can only diminish the benefit of synaptic caching.

If, on the other hand, there is a late-phase maintenance cost during learning, postponing consolidation will reduce costs further (see Results subsection, “Synaptic caching”).

These arguments only consider the cost of learning and not the cost of computing once the network has learned. We are currently studying this much more complicated situation. We have shown earlier that learning rules can be designed to reduce costs associated to computation (Sacramento et al., 2015), but the precise interaction of computation cost and plasticity cost is not clear at the moment.

4) Please test for comparison the energy consumption of other learning rules performing perceptron learning (e.g., D’Souza et al., 2010).

We have now implemented the learning rule from D’Souza et al., 2010, and find that indeed large energy savings can still be achieved (see new Figure 2—figure supplement 1), showing that the synaptic caching principle is general and does not depend on implementation details.